# Mucin Expression Profiles in Ulcerative Colitis: New Insights on the Histological Mucosal Healing

**DOI:** 10.3390/ijms25031858

**Published:** 2024-02-03

**Authors:** Giuseppe Leoncini, Luigi Cari, Simona Ronchetti, Francesco Donato, Laura Caruso, Cristina Calafà, Vincenzo Villanacci

**Affiliations:** 1First Pathology Division, Department of Pathology and Laboratory Medicine, Fondazione IRCCS Istituto Nazionale dei Tumori, 20133 Milan, Italy; giuseppe.leoncini@istitutotumori.mi.it; 2Pharmacology Division, Department of Medicine and Surgery, University of Perugia, 06132 Perugia, Italy; 3Unit of Hygiene, Epidemiology and Public Health, University of Brescia, 25123 Brescia, Italy; 4Pathology Unit, Department of Pathology and Laboratory Medicine, Desenzano del Garda Hospital, ASST del Garda, 25015 Brescia, Italy; 5Institute of Pathology, ASST Spedali Civili, 25123 Brescia, Italy

**Keywords:** ulcerative colitis, mucosal healing, mucins, mucus barrier, mucin enhancer

## Abstract

A structural weakness of the mucus barrier (MB) is thought to be a cause of ulcerative colitis (UC). This study aims to investigate the mucin (MUC) composition of MB in normal mucosa and UC. Ileocolonic biopsies were taken at disease onset and after treatment in 40 patients, including 20 with relapsing and 20 with remitting UC. Ileocolonic biopsies from 10 non-IBD patients were included as controls. Gut-specific MUC1, MUC2, MUC4, MUC5B, MUC12, MUC13, MUC15, and MUC17 were evaluated immunohistochemically. The promoters of mucin genes were also examined. Normal mucosa showed MUC2, MUC5B, and MUC13 in terminal ileum and colon, MUC17 in ileum, and MUC1, MUC4, MUC12, and MUC15 in colon. Membranous, cytoplasmic and vacuolar expressions were highlighted. Overall, the mucin expression was abnormal in UC. Derangements in MUC1, MUC4, and MUC5B were detected both at onset and after treatment. MUC2 and MUC13 were unaffected. Sequence analysis revealed glucocorticoid-responsive elements in the *MUC1* promoter, retinoic-acid-responsive elements in the *MUC4* promoter, and butyrate-responsive elements in the *MUC5B* promoter. In conclusion, MUCs exhibited distinct expression patterns in the gut. Their expression was disrupted in UC, regardless of the treatment protocols. Abnormal MUC1, MUC4, and MUC5B expression marked the barrier dysfunction in UC.

## 1. Introduction

Ulcerative colitis (UC) is an inflammatory bowel disease (IBD) of the colorectal mucosa; it has a multifaceted etiology in which host and environmental factors play important roles [1]. Among the host factors, mucus abnormalities have recently gained attention as causative co-agents of UC [2]. Indeed, an altered mucus layer, dysbiosis, and abnormal mucus–microbiota interactions have been established to predispose to IBD [3]. 

Mucus forms a hydrogel that protects mucous membranes. The main protein components of mucus are the mucins (MUCs), which are high-molecular-weight glycoproteins expressed by epithelial cells. A total of 22 mucins have been identified to date, and they are classified into two main categories [4,5,6]. Membrane-bound mucins have a transmembrane domain that anchors them to the plasma membrane [7]. Secretory mucins are delivered into the lumen, regulating the rheological properties of the mucus layer [8]. Among them, MUC7 is the only one to be secreted by salivary glands [9]. In the gut, mucins are mainly produced by goblet cells that result from the activation of different pathways for lineage specification [10]. 

Mucus forms a barrier that protects the gut epithelial lining from injuries and controls the resident flora. Normal mucus–microbiota interplay contributes to gut homeostasis [3]. Altered mucin production is able to weaken the mucus barrier (MB), triggering an immune response and predisposing to UC onset [11,12,13,14]. From a histological perspective, the finding of mucin depletion in the mucosal samples from UC patients points toward perturbations in mucin production, as confirmed by mucin gene expression profiling [15,16]. Impairment of MUC2 expression has been reported to be a major cause of mucosal injury in IBD [11,17,18,19].

Mucosal healing and deep remission are the main goals of the clinical management of IBD. Attempts to achieve mucosal healing by immunosuppression have only produced transient symptomatic control to date. In addition to the conventional first-line treatments, consisting of mesalamine (5-ASA), glucocorticoids, and immunosuppressants (e.g., azathioprine, methotrexate), novel drugs have been developed and employed in recent decades, including monoclonal antibodies (e.g., infliximab, adalimumab, vedolizumab, and ustekinumab). Moreover, the small-molecule drug tofacitinib (a Janus kinase inhibitor) has recently been employed [20]. However, side effects from long-term immunosuppressive and anti-inflammatory treatments, including an increased risk of infections [21] and hematological malignancies [22], have been reported. Therefore, the identification of novel treatment strategies is required. Because the restoration of normal mucin expression might facilitate mucosal healing, the roles of these glycoproteins in the gut should be investigated more thoroughly. The present study aims to provide a wide mucin expression landscape in UC mucosal samples, as compared to normal gut mucosa. Then, promoter sequences of altered mucin genes are examined to obtain preliminary insights into how these genes are regulated and which are suitable for consideration from a novel treatment perspective. 

## 2. Results

The study considered 40 patients with a histologically confirmed diagnosis of UC. Before treatment, all UC patients complained of abdominal pain, bloody diarrhea, and mucus discharge, and some also had fever and rectal urgency. Endoscopic findings showed a Mayo score ranging from 1 to 3. UC patients had completed first-line treatment with mesalamine, glucocorticoids (with or without NSAIDs), infliximab, or adalimumab before undergoing a second ileocolonic biopsy at a median interval of 56 days. By design, 20 of these patients had relapsing UC and the other 20 had remitting UC. The study also included 10 control patients who had undergone ileocolonoscopy for aspecific symptoms (e.g., abdominal pain, watery diarrhea) without clinical or histological evidence of UC (Table 1).

Histological findings leading to a diagnosis of UC included architectural distortion and the presence of immune cells in the lamina propria, including plasma cells and eosinophils. Histological descriptors of active disease, including scattered neutrophils, glandular injuries such as cryptitis and crypt abscesses, mucosal erosions, and ulcerations, were also detected. Control patients had normal glandular architecture and scattered immune cells in the mucosa, without basal plasma cells or neutrophil infiltration. Epithelial injuries were not detected.

### 2.1. Normal Ileal and Colonic Mucosa Exhibits Different Mucin Expression Profiles

First, to establish the mucin expression pattern in normal gut mucosa, we examined biopsy specimens from the 10 control patients (Figure 1). This analysis revealed identical patterns in all samples. Three mucins, MUC2, MUC5B, and MUC13, were expressed in both the terminal ileum and the colon. MUC17 was expressed only in ileal mucosa, while MUC1, MUC4, MUC12, and MUC15 were found only in colonic mucosa. Due to the faint membranous staining of MUC15 in IHC, its expression was also assessed by immunofluorescence (Appendix A). From these results, we defined MUC17 as ileum-restricted mucin and MUC1, MUC4, MUC12, and MUC15 as colon-restricted mucins. At the sub-cellular level, a single staining pattern was detected for MUC2 (vacuolar), MUC13 (cytoplasmic, in the colon), and MUC12, MUC15, and MUC17 (all membranous). Instead, a combined pattern of expression was observed for MUC1 (vacuolar and membranous), MUC13 (cytoplasmic and membranous, in the ileum), and MUC4 and MUC5B (both cytoplasmic and vacuolar). These results show that mucin expression differs between the ileum and colon in control patients. The mucin expression pattern in the normal gut mucosa of each patient is reported in Appendix A. 

### 2.2. Mucin Expression Profiles Show Both Inter-Site and Intra-Site Abnormalities in Treatment-Naïve UC Patients, Highlighting MUC1-MUC4-MUC5B Molecular Signature

We compared the colonic mucin expression pattern between specimens from control and UC patients. Moreover, in UC patients, the mucin expression pattern was also compared between active and quiescent disease (Figure 2). The results showed similar expression patterns of MUC2, MUC12, and MUC13 in treatment-naïve UC patients, as compared to control patients. Several intra-site abnormalities were detected in the colonic mucin expression, mostly affecting MUC1, MUC4, and MUC5B (Figure 3; Table 2). In all 40 patients, MUC1 showed a gain in cytoplasmic expression (*p* < 0.0001), whereas MUC4 retained its cytoplasmic expression but lost its vacuolar labeling (*p* < 0.0001), despite preserved goblet cell morphology upon histological analysis. Loss of vacuolar labeling was also observed for MUC5B, but only in 25 patients (62.5%, *p* = 0.0002). Although the lack of significance from a statistical point of view, it is worth noting that MUC15 expression was not detected in 17 cases (42.5%). On the other hand, the ileum-restricted MUC17 was expressed in colonic mucosa in 7 UC cases (17.5%), but with a cytoplasmic staining pattern instead of the membranous pattern seen in the ileum. The colonic expression of MUC17 in a subset of patients represents an inter-site (segmental) mucin abnormality, affecting the MB composition. Moreover, the transcriptional assessment revealed abnormal expression profiles for MUC1, MUC5B, MUC12, MUC13, and MUC15 in UC patients (Appendix A).

Overall, these results suggest that the mucosal inflammation of UC affects the expression of some, but not all, gastrointestinal mucins. The mucin expression pattern in the gut mucosa of each treatment-naïve UC patient is reported in Appendix A.

### 2.3. MUC1-MUC4-MUC5B Derangement Is Also Detectable in Treated UC Patients

After first-line treatment, the colonic mucin expression patterns slightly differed between the two subgroups (relapsing and remitting) of UC patients (Figure 4; Table 2). In the 20 patients with relapsing UC, the mucin expression abnormalities mirrored those in treatment-naïve patients, except for the vacuolar expression of MUC1, which is present in a lower number of relapsing patients compared to treatment-naïve UC patients (30% vs. 85%) (Appendix A). In the 20 patients with remitting UC, differences were seen for MUC1, which normalized in the six cases treated with glucocorticoids but retained an overall abnormal profile. MUC4 expression remained abnormal. MUC5B was abnormal in eight treated patients with relapsing disease (20%) and ten patients with remitting disease (25%). The mucin expression pattern in the gut mucosa of each relapsing and remitting UC patient is reported in Appendix A.

Overall, our data suggest that a broad spectrum of mucin derangement occurs in UC, involving MUC1, MUC4, and MUC5B. These results indicate that the normal mucin expression profile was not restored with immunosuppressant administration in either relapsing or remitting UC. Moreover, treated patients did not show complete mucin restoration, despite their transient symptomatic relief.

### 2.4. Mucins Are Druggable Targets

The promoter analysis of MUC1, MUC4, and MUC5B genes showed the potential involvement of specific transcription factors that can regulate mucin expression under pharmacological treatments or natural compounds administration. Particularly, MUC1 (accession number X69118.1) promoter analysis showed several putative binding sites for Glucocorticoid Receptor alpha (GRα), suggesting MUC1 to be a direct glucocorticoid-responsive gene (Appendix A). The promoter region of MUC4 (accession number AF241535.2) showed putative binding sites for retinoic acid receptor α (RARα), RARβ, or the heterodimer RARβ/RXRα, through which ATRA exerts its pleiotropic effects, suggesting a direct regulation of MUC4 expression by retinoids (Appendix A). Our analysis also showed the MUC5B promoter region to have putative binding sites for butyrate. Butyrate is a short-chain fatty acid (SCFA) produced by gut microbiota, exerting modulatory activities. Five putative butyrate response elements (BRE) were found in the MUC5B promoter that could represent candidate sequences for the butyrate-mediated regulation of MUC5B transcription (Appendix A).

## 3. Discussion

Intestinal mucins are well established to mainly be produced by goblet cells. The present study shows that mucin protein expression pattern differs in the normal gut between the ileum and the colon. Both quantitative (goblet cell reduction [23]) and qualitative abnormalities (mucin depletion [24]) have been described in UC patients. MUC2, MUC5B, and MUC13 were expressed in small and large bowels and could be considered a gut mucin core. In contrast, MUC17 was found only in the ileum, whereas MUC1, MUC4, MUC12, and MUC15 were restricted to the colon. In UC patients, both goblet cell reduction and mucin depletion were observed. Moreover, our comprehensive evaluation of gut-specific mucins showed a wide range of mucin expression abnormalities, mostly consisting of changes in both the intestinal segment and the subcellular compartment (intra-site abnormalities). The colonic expression of MUC17 in a subset of patients represents an inter-site abnormality, affecting the MB composition through the introduction of an ileum-restricted mucin to the colonic microenvironment. 

Among the colonic mucins, the most affected were MUC1, MUC4, and MUC5B, in both active and quiescent disease, indicating the occurrence of an abnormal sub-cellular expression within the expected segment, rather than a segmental shift in expression (intra-site abnormalities). 

MUC1 was expressed in a combined vacuolar and membranous pattern in normal colonic mucosa. In active UC, there was a gain in cytoplasmic MUC1 expression. This finding is in line with a report of an increase in *MUC1* mRNA expression in UC in relation to disease severity [19]. In treated patients, MUC1 expression was restored in six patients treated with glucocorticoids, leading to remission. Intriguingly, we identified putative glucocorticoid-responsive elements in the promoter of *MUC1*. 

MUC4 expression in normal colonic mucosa showed a combined cytoplasmic and vacuolar pattern. It was reduced in both active and quiescent UC, with the loss of vacuolar labeling and preserved goblet cell morphology. 

MUC5B was also expressed in a combined cytoplasmic and vacuolar pattern in normal colonic mucosa. In UC, about half of the patients had aberrant MUC5B expression in terms of the loss of vacuolar labeling. Perturbations in MUC5B expression have previously been correlated with disease severity in UC [25]. Moreover, MUC5B downregulation was found to affect gastrointestinal carcinogenesis, due to the involvement of the Wnt/β-catenin pathway [26]. Minor abnormalities were also detected in other MUC types. In almost half of UC cases, there was a loss of membranous MUC15 immunolabeling. In seven cases, MUC17 was unexpectedly found in colonic mucosa with a cytoplasmic pattern. In contrast, MUC12 and MUC13 were not affected by UC and they retained normal expression patterns. Our findings regarding MUC12 protein expression are in contrast with the reported downregulation of *MUC12* mRNA in IBD [27].

The derangements we identified in the expression of MUC1, MUC4, and MUC5B mark the MB dysfunction in UC. Thus, such abnormalities can be considered as a molecular signature of UC, a sort of timed device for clinical relapse that immunosuppressive agents alone cannot defuse. 

Therapy that enhances or restores mucin expression has not been proposed to date. We speculate that mucin enhancement could provide long-lasting symptomatic control of UC by counteracting the weakness of the mucosal barrier. Since mucin expression abnormalities are thought to be the earliest cause of UC [11], mucin restoration should not be excluded from the definition of mucosal healing. In the UC, mucosal healing is currently defined, from the endoscopic perspective, as the disappearance of mucosal erosions and ulcers or as the disappearance of inflammatory and ulcerative lesions [28,29]. From the histological perspective, mucosal healing is defined by the absence of active mucosal inflammation, i.e., neutrophil infiltrates in the *lamina propria* [28,29,30]. However, disagreement persists regarding both the terminology and clinical–pathological correlations of mucosal healing. The endoscopic definition does not correlate with the histological definition since histological healing cannot be inferred from endoscopic features [31]. We should not define mucosal healing without considering the histological features and the mucin expression dynamics, including derangement and restoration. The peculiar MUC1-MUC4-MUC5B signature could highlight a mucosal barrier disruption and should be evaluated along with the histological features, mostly in treated patients. Such a simple assessment could help in planning the clinical management after first-line therapy in UC via the identification of a subset of patients that need different therapies. 

Mucins could be considered as druggable targets. The responsiveness of *MUC1* to glucocorticoid-based therapy confirms the direct role played by glucocorticoids in directly regulating the *MUC1* promoter. Therefore, the transcriptional regulation of the *MUC1* promoter by glucocorticoids could help to achieve mucin restoration in UC. Retinoic acid, a vitamin A metabolite involved in several biological processes, including inflammation, was found to potentially transcriptionally regulate *MUC4* expression [32]. Interestingly, it was previously demonstrated that the administration of an encapsulated all-trans isomer of retinoic acid (ATRA) improved the symptoms of colitis [33] and maintained immune tolerance in IBD animal models [34,35,36]. Therefore, ATRA could represent a candidate transcriptional regulator of the *MUC4* gene, restoring the MUC4 levels in the MB. The short-chain fatty acid (SCFA) butyrate was reported to be beneficial for mucin normalization in IBD [37,38,39,40,41], as well as improving the gut permeability by regulating the tight junction assembly [42,43]. The presence of putative BREs in the *MUC5B* promoter suggests that butyrate could play a role in the transcriptional regulation of MUC5B secretion toward normalization of the mucus composition in UC. Further studies are needed to confirm the putative role played by the proposed mucin enhancers in restoring the qualities of the MB in UC patients.

Our achievements have some limitations, including the retrospective design of the study and the relatively small number of patients.

In conclusion, our data showed the following: (i) MB composition encompassed a broad variety of mucins in the normal gut mucosa, with site-specific (segmental) differences; (ii) a permanent derangement in the mucin expression profile characterized UC patients, regardless of both the treatment protocol and the clinical outcomes; (iii) the available immunosuppressive drugs are not able to normalize the mucin expression profile, particularly MUC1, MUC4, and MUC5B, even in patients with remitting UC. From a diagnostic perspective, the complete restoration of mucin expression should be considered a histological marker of mucosal healing in UC. From a therapeutic perspective, the identification of specific mucin impairments in treated patients could prompt the administration of selected mucin enhancers, promoting long-lasting clinical remission and positively affecting the patient’s quality of life. The histological assessment of mucin expression profile in treated patients could help define mucosal healing and should be considered as a diagnostic requirement in UC management, paving the way for personalized therapy.

## 4. Materials and Methods

### 4.1. Specimens and Clinical Data

The study used previously collected ileocolonic biopsy specimens and clinical data from 50 patients who presented to the gastroenterology departments of Spedali Civili and Ospedale di Desenzano del Garda (both in Brescia, Italy), after approval by the Ethics Committee of Spedali Civili, Brescia, Italy (study 4450/2020; approval number 4471). The study group included 40 patients with histologically confirmed UC and 10 patients who had undergone ileocolonoscopy for nonspecific symptoms, but whose intestinal mucosa was found to be normal upon histology. UC patients were included if paired biopsy specimens were available, both from the onset and after the first-line treatment. For the purposes of this study, 20 patients with relapsing UC and 20 with remitting UC were selected. Patients with relapsing UC were those in which treatment provided only transient symptomatic relief, who had histological features consistent with active disease, and in whom the Mayo score [44] was ≥1. Cases with remitting UC were those with complete symptomatic control after treatment, bland mucosal inflammation, an absence of neutrophil infiltration or mucosal epithelial injuries at histology, and an endoscopic Mayo score of 0. Control patients were included only if they had no other intestinal disease, including neoplasms; for these patients, only single biopsy series were studied. All UC patients had undergone ileocolonoscopy the first time because of clinical evidence of colitis, and the original pathologist’s diagnosis (UC or not) was verified for this study by two investigators (GL and VV).

Endoscopic sampling included at least two pinch biopsies from each intestinal segment (terminal ileum, right colon, transverse colon, descending colon, sigmoid, and rectum). The obtained mucosal samples were put onto cellulose acetate filters in an anatomically correct orientation at the trans-sectional cut to minimize manipulation artifacts during later steps. 

For all patients we collected, from medical charts, data on age, sex, symptoms, and type of medical treatment. In addition, for UC patients we collected the Mayo score (range from 0 to 3) before and after first-line treatment, and the time between the first and second biopsies. 

### 4.2. Histological and Immunohistochemical Assessments

Biopsy specimens were formalin-fixed and paraffin-embedded (FFPE), and then cut sequentially into multilevel three-micron-thick slices. For histological analysis, they were stained with hematoxylin-eosin dyes. Immunohistochemical analysis focused on eight mucins (MUC1, MUC2, MUC4, MUC5B, MUC12, MUC13, MUC15, and MUC17), selected for their documented expression in the gut mucosa (http://www.proteinatlas.org (accessed on 30 January 2024)). The antibodies used in these analyses are provided in Appendix A. 

Immunohistochemical results were interpreted according to the intestinal segment (ileum vs. colon) and the sub-cellular compartment, as follows: *membranous* (m), when a mucin was expressed on the apical membrane of goblet cells; *cytoplasmic* (c), when it was expressed in the cytoplasm, but not vacuoles; and *vacuolar* (v), when it was found only in cytoplasmic vacuoles. Combinations of these patterns were also considered (Figure 5). Images were acquired by using a Zeiss Axio Scope A1 microscope (Zeiss, Oberkochen, Germany).

For mucin 15 (MUC15) immunofluorescence staining, sections (6 µm thick) were cut from FFPE biopsy specimens, deparaffinized, and rehydrated in two changes of xylene for 5 min each. The slides were dipped in 100% alcohol twice for 2 min, followed by rinses in 95%, 90%, and 70% alcohol, for 2 min each. Antigen retrieval was performed by dipping the slides in citrate buffer in boiling H_2_O for 90 min. After three washes in PBS, the slides were incubated in blocking buffer (0.1% TritonX-100, 1% bovine serum albumin [BSA], 10% horse serum in phosphate-buffered saline [PBS]) for 1 h, followed by overnight incubation with anti-MUC15 antibody (rabbit, BS-5878R, 1:100 dilution; citrate buffer antigen retrieval) (Invitrogen, Waltham, Massachusetts, USA). Slides were incubated after 3 washes in 0.1% Tween PBS for 1 h, with secondary anti-rabbit antibody (Alexafluor 488, 1:100 dilution) (Invitrogen). Sixty-six focal planes (Z-range 19.5 μm, Z-step 0.30 μm) were acquired by using a Nikon Eclipse Ti microscope equipped with Confocal Spinning Disc Crest X-Light V2 fluorescence system (Nikon Europe B.V., Amstelveen, The Netherlands). Maximum intensity projection (MIP), deconvolution, and confocal 3D reconstruction were carried out using the NIS-Elements AR software version 2.2 (Nikon Europe B.V., Amstelveen, The Netherlands). The primary antibody used in these analyses is presented in Appendix A.

### 4.3. Transcriptional Mucin Expression Profile

Transcriptomics data were downloaded from the *Inflammatory Bowel Disease Multi’omics Database* (IBDMDB) [45]. Mucin expression was analyzed in the colon–rectum bioptic samples from Healthy donors (HS, n = 29) and UC patients (UC, n = 47). 

### 4.4. Mucin Promoter Analysis

Promoter regions of mucin genes were analyzed to identify sites where specific molecular factors regulate gene expression. Putative transcription factor binding sites were identified using PROMO virtual laboratory version 8.3 (https://alggen.lsi.upc.es/cgi-bin/promo_v3/promo/promoinit.cgi?dirDB=TF_8.3 (accessed on 30 January 2024)) [46], which generated lists of putative binding sites. These lists were then visually scanned for sites of interest to mucin regulation in intestinal tissue. Butyrate response elements were searched using BLASTn (https://blast.ncbi.nlm.nih.gov/Blast.cgi (accessed on 30 January 2024)).

### 4.5. Statistical Analysis

In each patient subgroup, the expression of each mucin was assessed, and the percentage of patients exhibiting each expression pattern (*cytoplasmic*, *vacuolar*, and *membranous*) was calculated. The frequencies of the different expression profiles were then compared between the subsets of patients. The contingency test (Chi-square test) was employed to evaluate the statistical significance of agreement in mucin expression patterns.

For the transcriptomics data comparison, the Kolmogorov–Smirnov normality test was performed to analyze the distribution of data. *p*-values were calculated using the unpaired t test with Welch’s correction for normally distributed data and the Mann–Whitney test for data with skewed distribution. A statistical analysis was conducted using PRISM version 9.5.1 (GraphPad Software, Boston, MA, USA), with statistical significance set at *p* < 0.05.

## Figures and Tables

**Figure 1 ijms-25-01858-f001:**
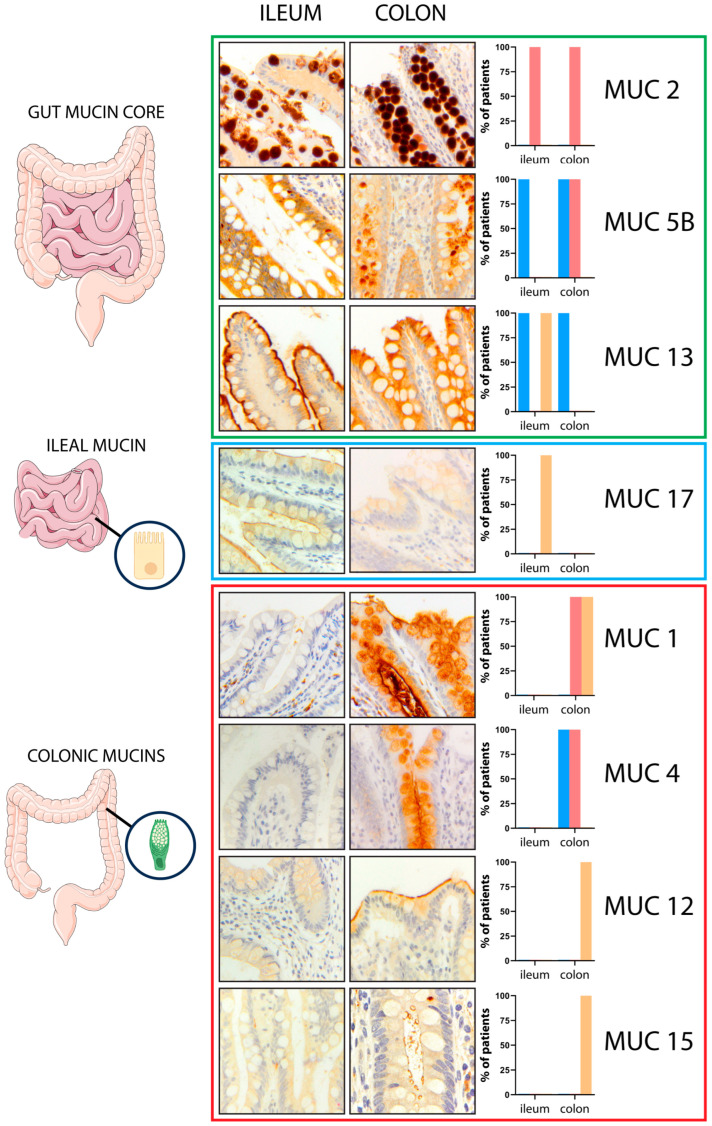
**Mucin expression across the gut.** MUC2, MUC5B, and MUC13 were diffusely expressed in both the terminal ileum and the colon (gut mucin core). MUC17 exhibited an ileal-restricted expression (ileal mucin), while MUC1, MUC4, MUC12, and MUC15 were found only in the colonic mucosa (colonic mucins). The graphs on the right represent the mucin expression pattern in the normal gut; the *cytoplasmic* (c) expression pattern is reported in blue bar, the *vacuolar* (v) expression pattern in red bar, and the *membranous* (m) expression pattern in yellow bar. Magnification ×20; magnification MUC15 right panel ×40.

**Figure 2 ijms-25-01858-f002:**
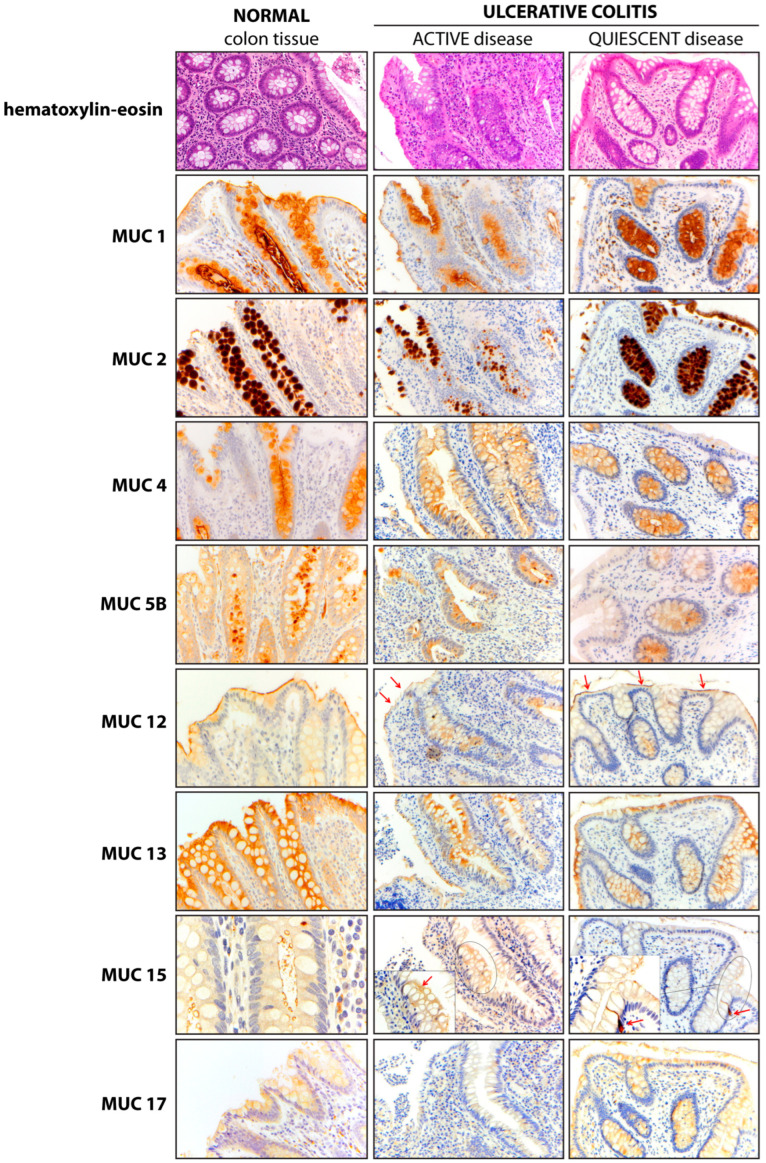
**Mucin expression pattern in normal and UC colonic mucosa.** The expression pattern of each mucin is presented in normal colonic mucosa (**left panel**) and in colonic mucosa from UC patients with active (**middle panel**) and quiescent (**right panel**) disease. In the MUC12 panel, red arrows mark the apical membranous (m) pattern. Meanwhile, in the MUC15 panel, an arrow points to a membranous (m) pattern in the active disease panel, and an MUC15-positive enteroendocrine cell is marked in the quiescent disease panel. Magnification ×20 (hematoxylin-eosin, MUC1, MUC2, MUC4, MUC5B, MUC12, MUC13, MUC15 central and right panels, and MUC17); Magnification ×40 MUC15 left panel and insert (central and right columns).

**Figure 3 ijms-25-01858-f003:**
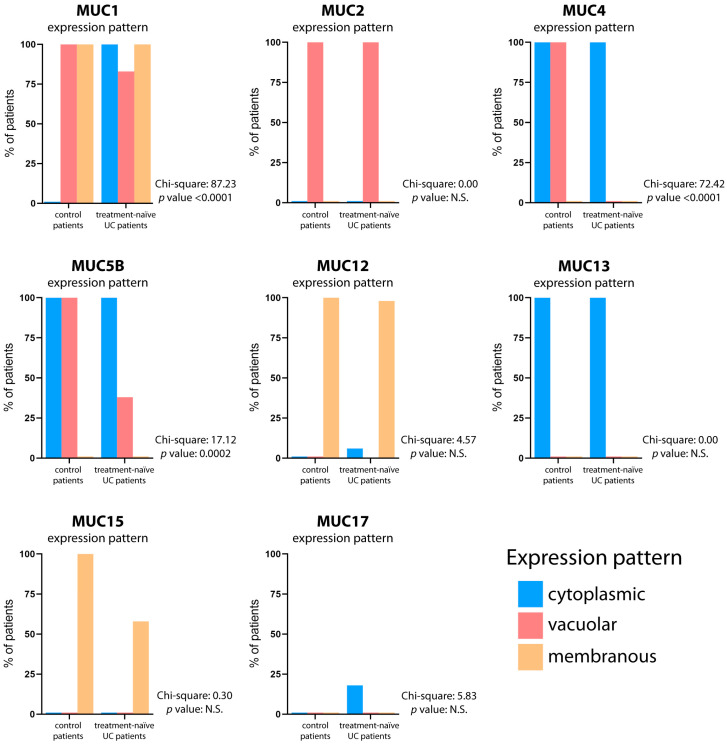
**Mucin expression pattern comparison between control and treatment-naïve UC patients.** The graphs depict the mucin expression patterns in both the normal gut and treatment-naïve UC patients. The percentage of patients displaying cytoplasmic (blue bar), vacuolar (red bar), and membranous (yellow bar) expression patterns is reported. The contingency test (Chi-square test) was utilized to assess the statistical significance of agreement in mucin expression patterns, with a significance level set at *p* < 0.05.

**Figure 4 ijms-25-01858-f004:**
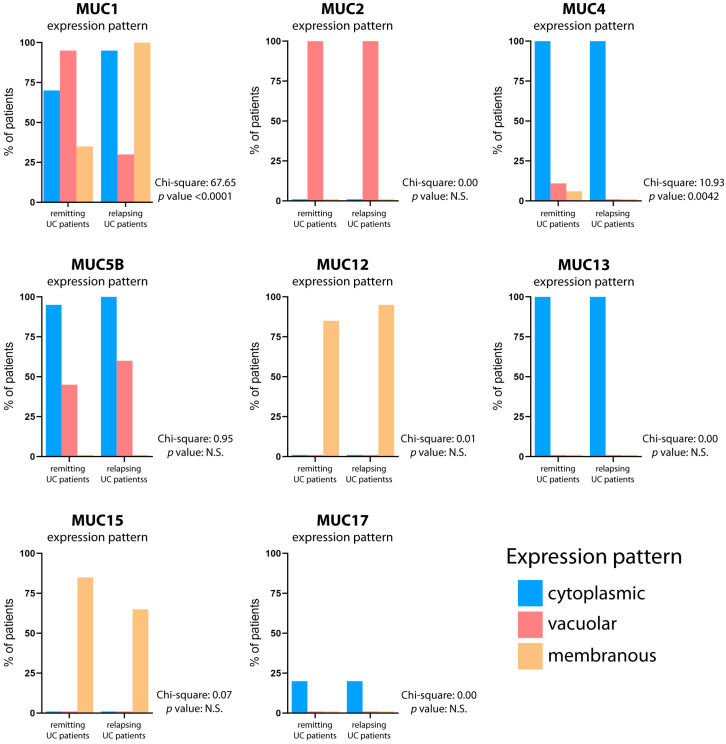
**Mucin expression pattern comparison between relapsing and remitting UC patients.** The graphs depict the mucin expression patterns in both the relapsing and remitting UC patients after first-line treatment. The percentage of patients displaying cytoplasmic (blue bar), vacuolar (red bar), and membranous (yellow bar) expression patterns is reported. The contingency test (Chi-square test) was utilized to assess the statistical significance of agreement in mucin expression patterns, with a significance level set at *p* < 0.05.

**Figure 5 ijms-25-01858-f005:**
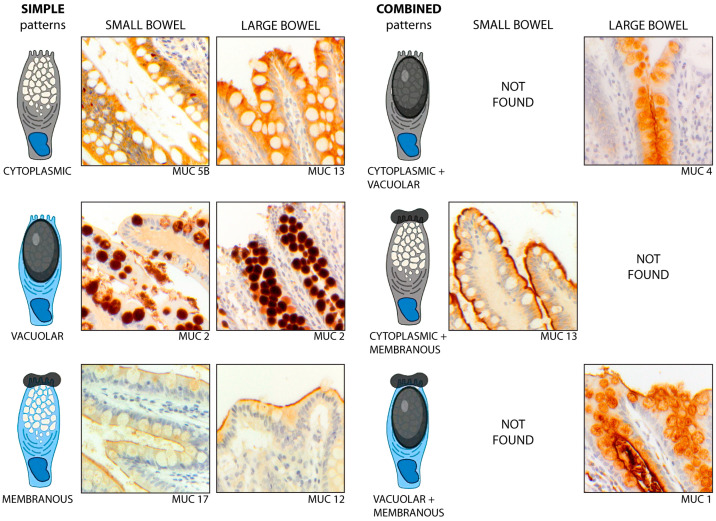
**Simple and combined patterns of mucin expression, according to the cellular compartment, in the non-IBD gut.** The simple patterns (**left**) include (1) the *cytoplasmic* (c) pattern, as exhibited by MUC5B in the ileum and by MUC13 in the colon; (2) the *vacuolar* (v) pattern, as shown by MUC2 in both segments; and (3) the *membranous* (m) pattern, exhibited by MUC12 in the colon and MUC17 in the ileum. The combined patterns (**right**) include (1) the *cytoplasmic-vacuolar* (c + v) pattern, as shown by MUC4 in the colon; (2) the *cytoplasmic-membranous* (c + m) pattern exhibited by MUC13 in the ileum; and (3) the *vacuolar-membranous* (v + m) pattern, as shown by MUC1 in the colon. Magnification ×20.

**Table 1 ijms-25-01858-t001:** Clinical characteristics of patients whose ileocolonic specimens were studied.

Characteristic		UC Cases (*n* = 40)			Controls (*n* = 10)
		Onset	Relapsing (*n* = 20)	Remitting (*n* = 20)	
Age, years ^a^		59 (16–72)	58 (16–68)	62 (24–72)	51 (28–67)
Sex, *n*					
	Male	17	6	11	5
	Female	23	14	9	5
Treatment, *n*					
	Mesalamine	N.A.	8	11	N.A.
	Glucocorticoids	N.A.	5	6	N.A.
	Infliximab or adalimumab	N.A.	7	3	N.A.
Interval between biopsies, days ^a^		56 (32–66)	48 (32–55)	61 (44–66)	N.A.
Mayo score, range		1–3	1–3	0	N.A.

N.A., not applicable. ^a^ Values are median (range).

**Table 2 ijms-25-01858-t002:** Mucin expression patterns in colon and in cellular compartments, in control tissue and ulcerative colitis. Expression in cellular compartments is classified as follows: c, cytoplasmic (non-vacuolar); m, membranous (apical membrane); and v, vacuolar (glycocalyx). (N.D., not detected/negative stain; N.R., not reactive, as expected).

Mucin	Controls (Large Bowel)	Ulcerative Colitis		
		UC Onset	Relapsing UC	Remitting UC
**MUC1**	v + m	c + v + m	c + m c + v + m	c + v + m v + m ^a^ c + m
**MUC2**	v	v	v	v
**MUC4**	c + v	c	c	c
**MUC5B**	c + v	c c + v	c c + v	c c + v
**MUC12**	m	m	m	m N.R.
**MUC13**	c	c	c	c
**MUC15**	m	m N.R.	m N.R.	m N.R.
**MUC17**	N.D.	c N.D.	c N.D.	c N.D.

^a^ Expression pattern in six cases at the end of glucocorticoid therapy.

## Data Availability

Analytic data and study material are available on reasonable request at First Pathology Division, Department of Pathology and Laboratory Medicine, Fondazione IRCCS Istituto Nazionale dei Tumori, Milan, Italy.

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
