# Peer review of "Mucin Expression Profiles in Ulcerative Colitis: New Insights on the Histological Mucosal Healing"

_ijms, 2024, doi:10.3390/ijms25031858_

Round 1
Reviewer 1 Report
Comments and Suggestions for Authors
Dr. Giuseppe and colleagues looked at mucin expression profiles in UC, both at histological relapse and mucosal healing phases. They took ileocolonic biopsies in 40 patients, 20 with relapsing and 20 with remitting UC, and from 10 healthy controls. They performed immunohistochemistry for 8 different MUCs and described the abnormal results. They also performed sequence analysis of the genes’ promoters. They concluded that MUC1, MUC4, and MUC5B expression marked the barrier dysfunction in UC.
The authors should be condemned of an important paper that describes important changes in the mucosal barrier in UC, and add information that might have an impact on future research.
Comments:
1. Page 1, line 46: I will be careful with such a statement, since the Gel-forming mucins are secreted, and may be called “secretory” in contrast to “membrane-bound”.
2. Page 2, line 48: Mucins may be secreted not only by the goblet cell but also by other epithelial cell types such as “mucus neck cells”.
3. Page 2, line 61: One should be very careful in giving NSAIDs to UC patients, since it may cause relapse.
4. Page 2, line 67: Usually in UC patients malignancy is caused by the disease.
5. Page 2, line 70: Mucin expression profile in UC has been studied before!
6. Page 2, Table 1: The average age of the UC patients is about 60 years, all completed “first-line treatment with…..” – are all newly diagnosed patients? Usually, UC is diagnosed in younger people. Please explain.
7. Page 2, Table 1: The controls are 10 years younger. Are you sure that there are no changes in MUCs expression with age?
8. Since only UC patients (and not Crohn’s disease patients) were examined, why the terminal ileum was biopsied? Too much information may confuse the reader.
9. Page 5, line 134: MUC17 appeared in UC colonic tissue, “switch” – was it disappeared from the ileum?
10. Table 2 is very important, giving an overview of MUCs distribution in UC patients versus control. I would skip the ileum.
Author Response
Dr. Giuseppe and colleagues looked at mucin expression profiles in UC, both at histological relapse and mucosal healing phases. They took ileocolonic biopsies in 40 patients, 20 with relapsing and 20 with remitting UC, and from 10 healthy controls. They performed immunohistochemistry for 8 different MUCs and described the abnormal results. They also performed sequence analysis of the genes’ promoters. They concluded that MUC1,MUC4, and MUC5B expression marked the barrier dysfunction in UC.
The authors should be condemned of an important paper that describes important changes in the mucosal barrier in UC, and add information that might have an impact on future research.
We would like to thank the Reviewer for finding our work interesting and for the helpful comments, you can find the point-to-point response below:
- (page1, line 46). I will be careful with such a statement, since the Gel-forming mucins are secreted, and may be called “secretory” in contrast to “membrane-bound”.
We agree with the Reviewer’s suggestion. The text has been corrected accordingly (lines 43-45).
- (page 2, line 48). Mucins may be secreted not only by goblet cells but also by other epithelial cell types, such as “mucus neck cells”.
We agree with the Reviewer’s suggestion, even though the mucin secretion is mostly sustained by goblet cells, according to the literature. The text has been corrected accordingly (line 46).
- (page 2, line 61). One should be very careful in giving NSAIDs to UC patients, since it may cause relapse.
Mesalamine (5’ASA) has been administered in the last years for mild disease at onset, based on the endoscopic Mayo Score, resulting in a transient symptomatic relief in a subset of patients. In that clinical setting, the ineffective symptomatic relief should not be intended as a causative agent of relapse (“since it may cause relapse”), since it can be due to the insufficient anti-inflammatory / immunosuppressive effect of Mesalamine. Glucocorticoids and monoclonal antibodies are also administered in that clinical setting. Nonetheless, we agree that the sentence can result as confounding by mentioning NSAIDs rather than Mesalamine. The text has been corrected accordingly (lines 58-61).
- (page 2, line 67). Usually in UC patients malignancy is caused by the disease.
We agree with the Reviewer’s suggestion, since long-term disease is known to be related to the increased risk of IBD-associated dysplasia and colorectal cancer. However, we are referring to hematological malignancies, as highlighted by the reference. Thus, the term hematological has been added into the text (line 65).
- (page 2, line 70). Mucin expression profile in UC has been studied before!
We agree with the Reviewer’s suggestion. Nonetheless, such expression profile has not been thoroughly studied so far, since it was focused on (or limited to) MUC2 expression in the most part of published studies. In our manuscript we provide, at the best of our knowledge, a comprehensive evaluation of the gut mucin expression in UC, including results from the assessment of 8 site-specific mucins in normal colon, treatment-naïve and treated patients. The text has been corrected accordingly (lines 68-69).
- (page 2, table 1). The average age of UC patients is about 60 years, all completed “first-line treatment with…”- are all newly diagnosed patients? Usually, UC is diagnosed in younger people. Please explain.
ALL the patients are included at first diagnosis, namely disease onset, and after the first-line treatment, as specified in the Methodology Section (4.1 Specimens and clinical data). The recruitment has been performed from the “real-life” clinical practice. In such a setting, were included very young people as well as cases of late-onset. UC can occur at any age (De Silva BC et al. World J Gastroenterol 2014) and the number of cases with late-onset is increasing (Kato H et al. World J Surg 2023), representing an emerging issue for the near future (Fries W et al Dig Liver Dis 2017).
- (page 2, table 1). The controls are 10 years younger. Are you that there are no changes in MUCs expression with age?
At the best of our knowledge, there are no age-related abnormalities in gut mucin secretion. Moreover, data about this topic were not found into the international literature.
- Since only UC patients (and not Crohn’s disease patients) were examined, why the terminal ileum is biopsied? Too much information may confuse the reader.
Ileal biopsies are included in the standard mapping protocol at the routinary endoscopy, in order to distinguish at the onset IBD vs non-IBD colitis and UC vs Crohn’s disease. Moreover, it should be noted that Crohn’s ileitis represents one of the clinical presentations of Crohn’s disease, that can also present as colitis or ileo-colitis, particularly in the case of late-onset. Again, the involvement of the right colon in UC can elicit reactive changes into the ileal mucosa (the so-called backwash ileitis), that should be differentiated by Crohn’s disease. For the mucin evaluation, our results showed MUC17 to be an ileal-restricted mucin, and the membranous pattern we detected for MUC17 was considered as a baseline to investigate the abnormalities of this mucin type in UC patients. Thus, excluding the ileal biopsies can compromise the diagnostic flow, as well as it can introduce a bias into the interpretation of MUC17 expression in UC.
- (page 5, line 134). MUC17 appeared in UC colonic tissue, “switch” – was it disappeared from the ileum?
We agree with the Reviewer’s suggestion, the term “switch” could result confusing. The text was corrected accordingly (lines 144-145).
- The table 2 is very important, giving an overview of the MUCs distribution in UC patients versus control. I would skip the ileum.
We agree with the Reviewer’s suggestion, the table was corrected accordingly.

Reviewer 2 Report
Comments and Suggestions for Authors
In this study, authors investigated mucin expression profiles in Ulcerative Colitis. They took Ileocolonic biopsies at disease onset and after treatment in 40 patients, including 20 with relapsing and 20 with remitting UC. Ileocolonic biopsies from 10 non-IBD patients were included as controls. Gut-specific MUC1, MUC2, MUC4, MUC5B, MUC12, MUC13, MUC15, and MUC17 were evaluated immunohistochemically. Promoters of mucin genes were also examined. They found that normal mucosa showed MUC2, MUC5B, and MUC13 in terminal ileum and colon, MUC17 in ileum, and MUC1, MUC4, MUC12, and MUC15 in colon. Membranous, cytoplasmic and vacuolar expressions were highlighted. Overall, the mucin expression was abnormal in UC. Derangements in MUC1, MUC4 and MUC5B were detected both at onset and after treatment. MUC2 and MUC13 were unaffected. Sequence analysis revealed glucocorticoid-responsive elements in the MUC1 promoter, retinoic acid-responsive elements in the MUC4 promoter, and butyrate-responsive elements in the MUC5B promoter. Finally, they made conclusions that MUCs exhibited distinct expression patterns in the gut. Their expression was disrupted in UC, regardless the treatment protocols. Abnormal MUC1, MUC4, and MUC5B expression marked the barrier dysfunction in UC. In general, this study is interesting. Here are some comments from this reviewer:
1. Mucins are secreted by goblet cells, which should be also investigated in this study.
2. Are goblet cells also decreased in the diseased samples?
3. Antibodies information for mucins detections should be provided.
4. The mucin expression profiles should be shown in an unbiased way. Analysis of online database concerning normal or IBD biopsies is recommended.
Author Response
In this study, authors investigated mucin expression profiles in Ulcerative Colitis. They took Ileocolonic biopsies at disease onset and after treatment in 40 patients, including 20 with relapsing and20 with remitting UC. Ileocolonic biopsies from 10 non-IBD patients were included as controls. Gut-specific MUC1, MUC2,MUC4, MUC5B, MUC12, MUC13, MUC15, and MUC17 were evaluated immunohistochemically. Promoters of mucin genes were also examined. They found that normal mucosa showed MUC2, MUC5B, and MUC13 in terminal ileum and colon, MUC17 in ileum, and MUC1, MUC4, MUC12, and MUC15 in colon. Membranous, cytoplasmic and vacuolar expressions were highlighted. Overall, the mucin expression was abnormal in UC. Derangements in MUC1, MUC4 and MUC5B were detected both at onset and after treatment. MUC2 and MUC13 were unaffected. Sequence analysis revealed glucocorticoid-responsive elements in the MUC1 promoter, retinoic acid-responsive elements in the MUC4 promoter, and butyrate-responsive elements in the MUC5B promoter. Finally, they made conclusions that MUCs exhibited distinct expression patterns in the gut. Their expression was disrupted in UC, regardless the treatment protocols. Abnormal MUC1, MUC4, and MUC5B expression marked the barrier dysfunction in UC. In general, this study is interesting.
We would like to thank the Reviewer for finding our work interesting and for the helpful comments, you can find the point-to-point response below:
- Mucins are secreted by goblet cells, which should be also investigated in this study.
Goblet cells represent the main cell type involved in the mucin secretion in the gut. Accordingly, we have expanded the first part of the Discussion section introducing a brief description of the role played by goblet cells (lines 232-235).
- Are goblet cells also decreased in the diseased samples?
Goblet cells are known to undergo perturbations in number and mucin secretion in UC. We have provided a brief description of the topic (lines 232-235), with references 23 and 24.
- Antibodies information for mucin detection should be provided
Please find such information into the Table S4 of the manuscript, as reported in lines 358-359.
- The mucin expression profiles should be also shown in an unbiased way. Analysis of online database concerning normal and IBD biopsies is recommended.
We agree with the suggestion of the Reviewer, even though it should be noted that mucin abnormalities we have found in our series mostly consist in their expression into abnormal cellular compartments. Online databases do not provide such information, in line with the novelty of our results. However, a comprehensive overview of the mucin transcriptional landscape is provided into an additional supplementary figure with a caption (Figure S2), and reported in the revised manuscript (lines 146-147).
